# Molecular Determinants of West Nile Virus Virulence and Pathogenesis in Vertebrate and Invertebrate Hosts

**DOI:** 10.3390/ijms21239117

**Published:** 2020-11-30

**Authors:** Lise Fiacre, Nonito Pagès, Emmanuel Albina, Jennifer Richardson, Sylvie Lecollinet, Gaëlle Gonzalez

**Affiliations:** 1UMR 1161 Virology, ANSES, INRAE, ENVA, ANSES Animal Health Laboratory, EURL for Equine Diseases, 94704 Maisons-Alfort, France; lise.fiacre.ext@anses.fr (L.F.); jennifer.richardson@vet-alfort.fr (J.R.); gaelle.gonzalez@anses.fr (G.G.); 2CIRAD, UMR ASTRE, F-97170 Petit Bourg, Guadeloupe, France; nonito.pages@cirad.fr (N.P.); emmanuel.albina@cirad.fr (E.A.); 3ASTRE, University Montpellier, CIRAD, INRAE, F-34398 Montpellier, France

**Keywords:** West Nile virus, virulence, molecular determinants, vertebrate and invertebrate hosts

## Abstract

West Nile virus (WNV), like the dengue virus (DENV) and yellow fever virus (YFV), are major arboviruses belonging to the *Flavivirus* genus. WNV is emerging or endemic in many countries around the world, affecting humans and other vertebrates. Since 1999, it has been considered to be a major public and veterinary health problem, causing diverse pathologies, ranging from a mild febrile state to severe neurological damage and death. WNV is transmitted in a bird–mosquito–bird cycle, and can occasionally infect humans and horses, both highly susceptible to the virus but considered dead-end hosts. Many studies have investigated the molecular determinants of WNV virulence, mainly with the ultimate objective of guiding vaccine development. Several vaccines are used in horses in different parts of the world, but there are no licensed WNV vaccines for humans, suggesting the need for greater understanding of the molecular determinants of virulence and antigenicity in different hosts. Owing to technical and economic considerations, WNV virulence factors have essentially been studied in rodent models, and the results cannot always be transported to mosquito vectors or to avian hosts. In this review, the known molecular determinants of WNV virulence, according to invertebrate (mosquitoes) or vertebrate hosts (mammalian and avian), are presented and discussed. This overview will highlight the differences and similarities found between WNV hosts and models, to provide a foundation for the prediction and anticipation of WNV re-emergence and its risk of global spread.

## 1. Introduction

Since its first isolation in 1937 in the province of Uganda [1], West Nile virus (WNV) has been a leading cause of neuroinvasive diseases in humans [2]. Its spread across the world, and notably its outbreak in New York City in 1999 and subsequent rapid diffusion across the American continent, identified WNV as a major public health problem [3]. Indeed, this event is the most representative example of WNV emergence and pathogenicity. A highly virulent strain of WNV lineage 1a was first isolated in New York City during the summer of 1999. From 1999 to 2016, 7 million infections were reported, leading to 24,000 neuroinvasive cases and 2300 deaths [4]. The virus then spread rapidly across North, Central, and South America [5], causing severe neurological disorders and death in humans and horses and affecting wild bird populations, in particular those belonging to the order Passeriformes, like the American crow (*Corvus brachyrhynchos*), mainly in the USA and Canada [6,7,8]. 

WNV is endemic in Africa, Europe, the Middle East, west and central Asia, Oceania (subtype Kunjin), and, most recently, North America, and is spreading into Central and South America [9,10]. WNV is a member of the *Flaviviridae* family. Its genome, a positive single-stranded RNA (+ssRNA) of 11 kb, encodes three structural proteins (C for capsid, (pr)M for (pre)membrane, and E for envelope) and seven non-structural (NS) proteins (NS1, NS2A, NS2B, NS3, NS4A, NS4B, and NS5), which are generated from a single polyprotein cleaved by several viral and host proteases [11]. The three structural proteins play a role in virion entry and egress [12] while the NS proteins play a role in intracellular multiplication, virion assembly, and escape from host immunity [13].

Based on phylogenetic analysis, WNV is classified into eight lineages [14]. WNV lineages 1 and 2 are the most widely distributed [15]. WNV lineage 1 strains have been reported in many areas, including America, Africa, Europe, India, the Middle-East, Russia, and Australia, and are responsible for major human and equine outbreaks, particularly in Europe and North America [6,16,17]. Several autochthonous cases of encephalitis after WNV lineage 1 infection were reported in Africa, the Middle East, and Europe during the last decades [6,18,19]. WNV lineage 1 can be divided into three sublineages. Lineage 1a includes isolates from Africa, the Middle East, America, and Europe. Lineage 1b includes the Kunjin virus (KUN), which is a subtype of the West Nile virus present in Australasia, a part of Oceania. Finally, lineage 1c corresponds to isolates present in India [20]. WNV lineage 2 viruses were originally isolated from sub-Saharan Africa and Madagascar, causing mainly mild fevers with no impairment of the central nervous system in humans. However, in 2009–2010, neuroinvasive disorders were observed, especially in Greece, Hungary, and South Africa [21,22]. Unprecedented WNV transmission seasons and outbreaks reported in Europe in 2010, 2012, 2013, and 2015 were associated with the introduction and spread of WNV lineage 2 strains [10]. These were first detected in Hungary in 2004 [23] and subsequently spread to the eastern part of Austria in 2008 [24,25], to the south of Italy in 2011 [26], and to the Balkan peninsula, including Greece in 2010 [27] and Serbia, Croatia, and Bulgaria in 2012 [28], and more recently reached Spain in 2017 [29] and Germany in 2018 [30,31]. Another WNV lineage 2 strain, first detected in 2004 in Rostov Oblast in Southern Russia [32], has also been occasionally reported in Europe, such as in Romania in 2010 and in Greece in 2018 [28,33,34]. The majority of neuroinvasive disorders in humans from 2010 onwards were associated with infection by WNV lineage 2 strains. Concomitantly, neuroinvasive diseases were diagnosed in horses and in humans in South Africa. These were also associated with infection by lineage 2 strains of the WNV [35]. Before that, less virulent lineage 2 strains had already circulated in South Africa, but without neuroinvasive cases. 

Since its emergence in New York in 1999, intensive work has been performed on the critical viral proteins and host factors implicated in WNV virulence and immunopathogenesis [11,36,37,38]. WNV infection results in an intricate balance between viral pathogenesis and immune-system mediated control [39,40]. However, the immunopathogenesis of WNV remains poorly understood. Neuronal injury may be directly caused by viral infection as well as through leucocytes infiltration and host inflammatory responses. Innate, humoral, and T-cell mediated host defenses orchestrate the control of WNV infection. Type I interferons are essential for restricting WNV replication and neuroinvasion into the CNS, while humoral immunity is involved in peripheral viral clearance [41,42]. However, the in vivo dynamics of innate and adaptive immune responses that are associated with differential outcomes of WNV infection in humans remains largely unknown. WNV shaped several countermeasures to evade host defenses (with NS1, NS2A, NS4B, and NS5 contributing to WNV evasion from the innate immune system). Moreover, WNV proteins like NS3 may also directly contribute to virus-mediated cytotoxicity, for example, by triggering apoptosis of infected cells [43]. However, a differential impact of the WNV strains on tissue pathogenesis is not necessarily linked to their virulence in a given host. In our previous study conducted by Donadieu et al., differential lesions of WNV Is98 and Kunjin strains after intracerebral inoculation of mice bore no relationship to virulence [41]. Indeed, the more virulent strain (WNV Is98) appeared to induce the lowest level of apoptosis. This strongly suggested that neuronal death in different areas of the brain was not directly linked to WNV virulence. Neuronal dysfunction alone might possibly have determined the clinical outcome of the disease. The observed differences are presumed to be of genetic origin. Mutations in genes implicated in viral adsorption, such as the E gene, or in viral replication have been shown to modify the organ tropism [13] and attraction of inflammatory cells implicated in the immune response [44].

Approximately 80% of human WNV infections are asymptomatic. In most symptomatic cases (20%), patients display mild febrile illness, associated with myalgia, arthralgia, headache, fatigue, intestinal complications, maculopapular rash, or lymphadenopathy. Less than 1% of these develop severe neurologic complications, manifested by different pathologies such as acute flaccid paralysis, meningitis, encephalitis, or eye disorders [45]. Encephalitis is the most severe neurological form, which occasionally can be fatal, particularly in the elderly and immunocompromised persons [11,46].

WNV is an arbovirus that is amplified in a bird–mosquito–bird enzootic transmission cycle, with wild birds as the primary hosts [47]. Humans and other mammals are considered to be accidental or dead-end hosts (Figure 1). 

WNV has been detected in 65 different mosquito species and more than 326 species of birds. Transmission to mammals and especially to humans is mainly carried out by *Culex* spp. (*Cx* spp.) mosquitoes. In birds, WNV produces an acute infection typically lasting up to a week. The viremia is typically short (a few days); however, the level of viremia can substantially differ among species (high in some bird species while lower in mammalians). Nevertheless, the virus has been shown to persist for several weeks in some wild birds. Avian species vary in susceptibility to infection and severity of the disease (from asymptomatic to severe neurological signs and sudden death), with mortality ranging from 0% to 100%. Symptoms vary from weight loss, lethargy, loss of vision, to neurological signs, including loss of coordination, tilting of the head, tremors, weakness, and lethargy [48,49]. In birds, WNV invades the CNS and other organs such as the heart, liver, spleen, and kidney. Passerines are the most affected, with the highest viremia and mortality rates, whereas gallinaceous birds exhibit no mortality and morbidity and a very low viremia. 

Depending on the geographic location, vector species vary. Of note, for a given vector species, behavior and susceptibility to arbovirus infection can vary between geographically separated populations. In Africa and the Middle East, the main vector is *Cx. univittatus*, but other species—such as *Cx. poicilipes*, *Cx. neavei*, *Cx. perexiguus*, *Cx. decens*, *Aedes albocephalus*, or *Mimomyia* spp.—are also present. In Europe, the main vectors seem to be *Cx. pipiens*, *Cx. modestus*, *Cx. perexiguus*, and *Coquillettidia richiardii* [50]. *Aedes albopictus* is known to be a vector of other flaviviruses, including Dengue virus (DENV) [51]. A recent study performed by Akiner and colleagues (2019) in Turkey, found WNV by PCR in *Ae. albopictus* collected in the field [52]. However, WNV has not as yet been isolated from *Ae. albopictus* in Europe [53]. The vector competence of *Ae. albopictus* for WNV has been demonstrated in laboratory experiments [54], while *Ae. aegypti*, another mosquito species implicated in flavivirus transmission, is not competent for WNV [55]. In Asia, *Cx. quinquefasciatus*, *Cx. tritaeniorhynchus*, and *Cx. vishnui* predominate [47,56,57]. Finally, in the United States, it appears that the species involved in WNV transmission are *Cx. pipiens*, *Cx. quinquefasciatus*, and *Cx. tarsalis* [48]. In Australia, *Cx. annulirostris* is considered the most important vector of WNV KUN [58]. WNV KUN has been isolated from other Australian species, including *Ae. tremulus*, *Cx. australicus*, *Cx. squamosus*, *Ae. alternans*, *Ae. normanensis*, *Ae. vigilax*, *Anopheles amictus*, and *Cx. quinquefasciatus*, but the role of most of these species is likely to be secondary [59].

WNV needs to cross many barriers in vertebrate and invertebrate hosts during its transmission cycle. After being ingested with infected blood, the WNV must first infect the midgut of the mosquito. The virus then crosses the midgut barrier and diffuses in the hemolymph to other organs, including the salivary glands, whose infection is a prerequisite for WNV transmission to new susceptible vertebrate hosts (Figure 2). In vertebrates, WNV first replicates in the keratinocytes in the epidermis. Thanks to the expression of numerous Pathogen Recognition Receptors (PRRs), keratinocytes are involved in the sensing of Pathogen-Associated Molecular Patterns (PAMPs). WNV infection leads to inflammatory responses in the keratinocytes that are characterized by the induction of type I or type III IFNs or the secretion of cytokines like TNFα (tumor necrosis factor) or IL-6 [60]. At later stages, WNV is internalized by skin dendritic cells (DCs) or Langerhans cells. Then, infected DCs migrate to the lymph nodes, and WNV disseminates to other lymphoid organs, such as the spleen. Most virulent WNV strains are inherently neuroinvasive and neurovirulent. To reach the brain, the virus needs to cross the blood–brain barrier (BBB), which can occur by two main routes. The first involves axonal retrograde transport along the spinal cord, and the second blood vessel transport. In the second case, the virus can use three methods to enter the brain: (I) the “Trojan horse”, in which the virus is internalized into lymphocytes that are able to cross the BBB; (II) permeabilization of the BBB by matrix metalloproteinases (MMPs) after MIF (macrophage migration inhibitory factor) and TNF-α secretion by leukocytes [61,62,63,64]; and (III) transcellular passage through the BBB, probably influenced by protein E glycosylation [11,65]. CNS entry of the WNV is a multistep process involving different mechanisms, whose importance would need to be better characterized. Infection of brain microvascular endothelial cells (BMECs) are thought to occur first. BMEC infection leads to increased expression of adhesion proteins and production of chemokines, responsible for leukocyte recruitments at the BBB. MMP9 can also play a role in the infiltration of WNV-infected leukocytes though the BBB [66]. Both mechanisms can contribute to WNV crossing the BBB by the “Trojan horse” mechanism [67] (Figure 3).

Some genetic changes are favorable to WNV transmissibility, thereby facilitating expansion of the WNV [71,72]. It is well documented that natural WNV infections are derived from a genetically diverse population of viral genomes. Within an individual host, WNV exists as a genetically heterogeneous mixture of variants that differ from a consensus nucleotide sequence; this population structure is referred to as a viral quasispecies [72,73]. WNV populations within field mosquitoes are more genetically diverse than those found in naturally infected birds. This results from several bottlenecks that shape the virus divergence during the progress of infection in birds, including the host defenses, varying cellular environments in different tissues, and anatomic barriers, such as the BBB. These multiple selective pressures determine tissue tropism and virus pathogenesis. Moreover, Jerzak and colleagues serially passaged a genetically homogenous population of WNVs in either mosquitoes or birds, and observed greater intra-host genetic diversity in mosquitoes than does chickens [74] (Figure 4). It is noteworthy, however, that most avian studies have involved fowl in which the replication of WNV is poor.

This predisposition for high genetic variability explains why, despite the small size of the viral genome (about 11 kb), wide genetic variation within the WNV lineages is observed across the world. The relationship between genetic heterogeneity and viral pathogenicity is variable. Indeed, viral heterogeneity was associated with enhanced pathogenicity for the mumps (ourlien) virus and poliovirus, but with weaker pathogenicity for the hepatitis C virus. Of note, WNV virulence in mice was negatively correlated with WNV genetic diversity [75]. WNV studies have shown that molecular determinants of virulence identified in mammals were not systematically shared by avian hosts or mosquito vectors, highlighting the importance of studying the molecular determinants of virulence across a broad host range [76].

Understanding the impact of WNV genetic variations on virulence is key for evaluation and anticipation of pathogenicity, for development of prophylactics, and for prediction and timely response to new epizootics. Identification of critical genetic determinants may be carried out through reverse genetic approaches, site-directed mutagenesis, and sequencing. It has been known for years that the genome of positive-strand RNA viruses, like flaviviruses, can be infectious when introduced into susceptible cells by transfection [77]. On this basis, reverse genetic technology permits manipulation of viral genomes and analysis of consequent changes in viral pathogenesis, as well as virulence, cell entry, or viral replication. Development of new molecular tools like PCR or RT-PCR at the end of the 1980s [78] greatly facilitated reverse genetic approaches. However, the absence of commercial reverse genetics kits led scientists to develop their own protocols [79]. The earliest procedures required bacterial or yeast cultures. In the former case, an isolated viral RNA extract is first reverse transcribed into cDNA and then typically inserted into a plasmid vector containing a promoter recognized by a DNA-dependent RNA polymerase (e.g., T7 or SP6) and the HDR/SV40pA sequences. Bacterial cultures are then transformed with the vector containing the viral sequence for amplification. Bacteria are collected and the vector containing the viral sequence, called the infectious clone (IC), is purified. Depending on the promoter choice, further steps may be needed before transfection of permissive cells with the IC to generate infectious viruses. However, toxicity of the full-length flavivirus genome for bacterial hosts has often been noticed. Moreover, this technology is particularly laborious. For this reason, new bacteria-free reverse genetic approaches for RNA viruses, and for flaviviruses in particular, have been developed since 1995, such as long PCR [80], circular polymerase extension cloning (CPEC), Gibson assembly [81,82], or the infectious subgenomic amplicons (ISA) [83] method. The ISA method’s main advantage is its rapidity. However, despite the use of a high-fidelity polymerase for PCR reactions, genomic heterogeneity was evidenced in the generated clones. In order to reduce the viral genomic heterogeneity, PCR fragments were cloned into plasmids in the subgenomic plasmids recombination method (SuPReME) (2019) [84] (Figure 5). These new protocols allow rapid generation of infectious clones and will support further investigation of the pathobiology of flaviviruses.

This review will present an overview of the known molecular determinants involved in WNV virulence and pathogenicity according to the host. The aim is to highlight the determinants that are specific to a given host, as well as those that the host have in common, in order to provide a better understanding of WNV virulence and to provide a foundation on which measures to control WNV spread and outbreaks can be proposed.

## 2. Mammalian Model

WNV infection has been extensively studied in mouse models. These studies have provided an understanding of viral pathology in humans and other susceptible mammalian hosts, and have shed light on the molecular pathways implicated in the innate control of infection in mammals and on molecular determinants of virulence. Researchers have mainly used C57BL/6 and C3H mouse models to decipher WNV infection and immunity [85,86]. In these inbred mice, inter-individual variation in the outcome of WNV infection is limited, thus facilitating interpretation and the inference of statistical significance. Outbred mice (NIH Swiss, CD-1 mouse model) have also been used, since they are more robust and provide a somewhat more physiological model, owing to their genetic heterogeneity. To enrich WNV studies and capture more phenotypic characteristics of West Nile disease, other strains of mice may be used, such as the Collaborative Cross (CC) strains. These include five inbred strains (C57BL/6J, A/J, 129S1/SvImJ, NOD/ShiLtJ, and NZO/H1LuJ) and three wild-type-derived strains (CAST/EiJ, PWK/PhJ, and WSB/EiJ) [87]. In conclusion, the mouse background strain should be carefully selected according to the objectives of the study in question. Of note, all regions of the WNV genome have been studied in mouse models. With the exception of the C and NS2B sequence regions, molecular determinants for mammalians have been identified in all parts of the WNV genome (Figure 6). These findings are detailed and discussed in the next sections.

### 2.1. The 5′UTR Non-Coding Region

The WNV polyprotein sequence is flanked by 5′ and 3′ untranslated regions (UTRs). The WNV 5′UTR region is 96 nt long and is highly conserved compared to other members of the flavivirus family. It contains two stem-loops (SLA and SLB). This region plays a role in viral cyclization and replication [89,90].

Audsley and colleagues [91] created chimeric viruses between an IC derived from the North American isolate WNV NY99 4132, isolated in 1999 in New York City, and the KUN virus, and more particularly between their 5′UTR and 3′UTR non-coding regions. They placed the non-coding regions of the IC NY99 into the KUN backbone, either the 3′UTR alone, the 5′UTR alone, or both of the non-coding regions of IC NY99. In vitro plaque assays performed on mammalian Vero and A549 cells and on mosquito C6/36 cells yielded similar results for the different mutants. Growth curve analysis identified minor differences between the mutant and the parental KUN strain. The main difference was obtained in vivo, upon intraperitoneal inoculation of mice with 10 PFU of the wild-type (wt) or mutant viruses. According to the results, the KUN virus with the 5′UTR of WNV NY99 (KUN-NY99 5′UTR) was significantly more virulent than the KUN wt. To understand which nucleotides of the NY99 5′UTR region were responsible for the increased virulence, the sequences of the 5′UTR region of KUN and NY99 were aligned. Only three nucleotide differences were found between the two strains. Indeed, while TTG is found at position 50–52 of the KUN virus, AAC is found in WNV NY99. Further alignment of multiple WNV strains showed that most WNVs have the sequence AAC. These nucleotides located in the 5′UTR region could partly explain the difference observed in virulence. However, when both non-coding regions of the KUN virus are replaced by those of NY99, the increased virulence is abolished. Thus, some component of the 3′UTR appeared to counterbalance the effect of the 5′UTR. In addition, replacement of the 5′UTR of NY99 by the KUN analogue failed to mitigate the virulence of NY99. Rather, this interchange resulted in only a marginal impact on virulence, probably because virulence depends upon multiple genetic determinants, as suggested by the +/− interplay between the 3′ and 5′UTR regions. However, taken together, these results suggest that the 5′UTR non-coding region of the WNV contributes to WNV virulence in mice.

### 2.2. Structural Proteins

#### 2.2.1. prM Protein

The glycoprotein precursor of the M protein, prM (26 kDa), is translocated into the ER by the C-terminal hydrophobic domain of the C protein [13]. The N-terminal region of the WNV prM contains one N-linked glycosylation site at amino acids 15–17, which plays a role in virus infectivity and particle release [92], and six conserved cysteine residues [93]. The prM protein promotes the correct folding of the E protein. Its major role is to allow the structural rearrangement of the E protein during transit through the secretory pathway, after prM cleavage by furin, to yield mature M proteins [94,95].

Hanna et al. [92] investigated the impact of prM and E glycosylation on WNV assembly and infectivity. Glycosylation of prM was ablated by a prM-N15Q mutation. The authors performed infection of HEK-293 cells with the wt virus and the prM-N15Q mutant WNV. They demonstrated that the viral RNA content decreased when the cells were infected with the mutant virus, influencing later stages of the virus life cycle in infected cells. We hypothesize that such a mutation would affect virus replication and dissemination in vivo.

Some studies have shown that mutations in the protein M could produce attenuated flaviviruses, such as JEV [96]. A recent study of Basset et al. [97] investigated the effect of protein M mutation in WNV. The authors generated WNV mutants by PCR from an IC of WNV Israel 98 (IS98) [3], which is genetically closely related to the NY99 strain. The first attenuated mutant (M-I36F) proved unstable. Introduction of a second mutation, A43G, created the stable mutant M-I36F/A43G. Smaller plaques were obtained on Vero cells for the M-I36F WNV, suggesting an attenuated phenotype, while normal plaques were seen for the M-I36F/A43G strain. Viral titers of both mutants (M-I36F and M-I36F/A43G), however, were considerably lower than those of IC IS98 or M-A43G. In vivo experiments performed on BALB/c mice showed that inoculation of wt or M-A43G WNV generated high mortality, whereas all mice infected by the M-I36F/A43G mutant survived. It is known that the M-36 residue is located in the pro-apoptotic domain of the M protein (« apoptoM ») and plays a role in the virulence of other flaviviruses, such as DENV [98,99]. The results obtained for WNV support this observation and show that M-36 mutants could be useful for vaccine development against WNV and other flaviviruses.

#### 2.2.2. E Protein

E protein is composed of 500 amino acid (aa) residues. Its final molecular weight depends on the glycosylation state of the protein, and ranges from 53 to 60 kDa. Its main role lies in the recognition of mammalian cell receptors as well as membrane fusion with the membrane of endocytic vesicles. It constitutes three domains, DI to DIII [100]. DI forms an eight-stranded β-barrel; DII is a long finger-like domain that contains at its tip the putative fusion peptide, which triggers fusion with the target cell membrane; and DIII assumes an immunoglobulin-like domain and is involved in receptor binding [101]. Several studies report the importance of protein E in WNV virulence, and in particular, its N-glycosylation site located at residues 154 to 156 [102,103].

Genetic comparison of all flaviviruses shows that there is a great diversity in the sequence of the gene encoding protein E. In 1995, a study focused on the amino acids NYS (positions 154–156) and their role in the glycosylation and antigenicity of protein E of the Kunjin virus, WNV lineage 1b [104]. In 1997, construction of a phylogenetic tree of WNV strains based on the sequence encoding the E protein evidenced differences in amino acids 154 to 157 for many strains of WNV [105]. Many subsequent studies have built upon these findings. 

Beasley and colleagues investigated the difference in the E protein of the highly virulent WNV lineage 1 NY99 strain and the less virulent Old World ETH76a strain, the objective being to understand why the WNV NY99 strain was more virulent. Sequence alignment revealed that the differences included five amino acids of the E protein, which might affect protein glycosylation. Site-directed mutagenesis and construction of chimeric viruses showed that the Asp residue at position 154 in NY99 results in a glycosylated E protein, whereas a Ser154 residue in ETH76a leads to a non-glycosylated E protein [102,103]. This may partly explain why virulence differs between these two strains.

Many studies concerning residues 154 to 156 have further confirmed the role of the glycosylated E protein of the WNV [102,106,107,108] in neurovirulence and neuroinvasion. Alsaleh et al. investigated the determinants of virulence for European-Mediterranean WNV strains. Chimeras were generated between the highly virulent IS98 strain and the non-pathogenic Malaysian Kunjin virus (KJMP-502) and evaluated in BALB/c mice. In contrast to WNV strain IS98, the KJMP-502 strain and all of the chimeras thereof possessing the KJMP-502 structural proteins were not neuroinvasive. Because the E protein is involved in receptor binding and cell entry, the authors generated the following chimeric viruses involving swaps of the E protein: IS98/E-KJMP and the reciprocal KJMP/E-IS98. The fatality rate was high (60%) 16 days after inoculation of mice with KJMP/E-IS98, while all mice challenged with IS98 died before the 10th day of infection and all mice challenged with IS98/E-KJMP remained healthy and survived for at least 18 days. This result suggests that even if the E protein plays a major role in the virulence of IS98, other factors may account for the residual difference in virulence between IS98 and KJMP/E-IS98. Other analyses showed that IS98 possesses the N-glycosylation site NYS at positions 154–156, whereas KJMP-502 has a proline residue at position 156, thus abrogating the N-glycosylation site. To understand whether the N-glycosylation site in the E protein is implicated in the neuroinvasiveness of IS98, the same authors employed site-directed mutagenesis to generate the IS98-E-S156P (−glyE) and its reciprocal mutant KJMP-E-P156S (+glyE). In vivo experiments showed that 70% of mice inoculated with IS98-E-S156P survived whereas all mice inoculated with IS98 died. Thirty percent of mice inoculated with KJMP-E-P156S died several days after inoculation whereas all mice inoculated with the parental KJMP-502 survived. These results suggested that although E-glycosylation is important for WNV pathogenicity, additional motifs in the E protein are probably involved in neuroinvasiveness and neurovirulence.

In the early 2000s, the distribution of WNV expanded to a greater extent in America than ever observed elsewhere, suggesting the possible emergence of a higher virulent phenotype. Davis et al. [109] sequenced the prM and E proteins of 74 isolates and the complete genome of 25 isolates obtained between 2001 and 2004. They found that genetic variants had arisen during the study period and were grouped temporally and geographically, suggesting that a dominant variant had rapidly emerged in North America. They also found that the E-V159A mutation (compared to initially introduced WNV NY99) was conserved in many WNV isolates recovered after 2001, suggesting a link with the enhanced WNV spread and pathology in America after the year 2000 [110].

A recent study by Kobayashi et al. [111] supports this scenario. WNV recombinants between the highly virulent lineage 1 strain NY99 (V159) and the less virulent Eg101 (I159) strain were generated but did not show any significant differences in in vitro cultures or in C57BL/6 mice inoculated intraperitoneally. In contrast, following the intracranial inoculation of mice, viral replication in the brain was higher for EgCME-E-I159V than for the parental strain EgCME. The mutation also increased levels of CD3+ and CD8+ T cells in the brain, suggesting that residue 159 of the E protein modulates WNV pathogenicity by influencing both viral replication and T-cell recruitment [111].

Zhang et al. [112] generated mutations in the E protein of the WNV that are known to mitigate virulence in other flaviviruses. From an IC of NY99, they created mutants in three regions; namely, the fusion loop (L107F), the receptor-binding domain III (A316V), and a stem helix (K440R). Neuroinvasion and neurovirulence of WNV NY99 IC-derived viruses and engineered variants encoding E protein substitutions were characterized in 3- to 4-week-old female NIH Swiss mice. The results showed that only the L107F mutation could reduce neuroinvasiveness without affecting neurovirulence.

Similarly, Kaiser et al. [113] studied a mutation in the E protein implicated in attenuated JEV phenotypes. JEV and WNV are closely related flaviviruses. Because the E-E138K mutation induced a decrease in JEV virulence, Kaiser et al. tested this mutation for WNV. In vitro studies performed on Vero and A549 cells showed that the multiplication kinetics of IC NY99 and the mutant E-E138K were similar. In vivo experiments in mice confirmed these results; that is, that the mutation did not decrease WNV virulence, whereas it did so for JEV. Indeed, the E-E138K mutation attenuated neither WNV neuroinvasion nor neurovirulence. This result underscores that differences in molecular virulence determinants exist between flaviviruses.

### 2.3. Non-Structural Proteins

#### 2.3.1. NS1 Protein

The NS1 glycoprotein is highly conserved among flaviviruses, with a molecular weight ranging from 46 to 55 kDa. NS1 has three N-glycosylation sites and exists as a monomer, a dimer (membrane-bound protein, mNS1), and a hexamer (secreted protein, sNS1). Its mature form is a homodimer [114], which is transported to the cell surface and acts as an immunomodulatory protein by decreasing the activity of the complement system [115]. NS1 plays a role in the modulation of host innate immunity and in viral replication. It is also a target for the generation of attenuated vaccines as well as a biomarker for viral diagnosis [116,117].

Flaviviruses share highly conserved NS1 N-glycosylation sites (N-X-S/T). Instead of presenting two glycosylation sites like other flaviviruses, WNV possesses three glycosylation sites on NS1 [116]. Whiteman et al. [116] wanted to develop an attenuated vaccine based on NS1 protein glycosylation mutants and used site-directed mutagenesis to generate the mutants NS1-N130A, NS1-N175A, and NS1-N207A. In parallel, a mutant lacking E and NS1 N-glycosylation sites was created (E-154S/NS1-N130A/NS1-N175A/NS1-N207A), based on evidence that E protein glycosylation decreases neurovirulence in mice [102]. The goal was to see whether the removal of glycosylation sites on both proteins further decreased the virulence. To this end, the in vitro infection kinetics of the NS1-130A/175A/207A and E-154S/NS1-130A/175A/207A mutants were compared with those of E-154S and the parental IC in Vero cells, P38801 mouse macrophages, and Neuro2A neuroblastoma cells. A number of differences were evidenced between the mutants and the parental IC in the three cell lines. Significant differences were observed between the IC and both E-154S and E-154S/NS1130A/175A/207A in Vero cells at 48 h, 72 h, and 96 h pi and between E-154S and E-154S/NS1-130A/175A/207A in P38801 macrophages at 72 h and 96 h pi. 

An in vivo study in NIH Swiss mice showed that, among the single glycosylation mutants, only the NS1-175A was significantly less neuroinvasive than the NY99 parental strain. Two other mutants, NS1-130A/207A and NS1-130A/175A/207A, were highly attenuated for neuroinvasion (with lethal dose 50 (LD50) 3000–50,000 fold higher than the parental NY99). The addition of the E-154S mutation (E-154S/NS1-130A/175A/207A) strongly decreased the neurovirulence and neuroinvasiveness. This study demonstrated that NS1 mutations did not affect the in vitro phenotype of the virus strains. In contrast, the higher the number of NS1 mutations, the greater was the impact on neuroinvasiveness and neurovirulence in mice. The combination of multiple NS1 mutations and the E-154S mutation (E-154S/NS1-130A/175A/207A) resulted in an attenuation for both neuroinvasiveness and neurovirulence [116].

WNV lineage 2 virulence determinants have been less investigated as these viruses only induced mild clinical signs in the past [118]. However, horses started developing encephalitis due to infection by WNV lineage 2 in 2009 [35]. A study conducted in 2016 focused on mutations implicated in WNV lineage 2 virulence, based on results obtained with the central European lineage 2 isolate 578/10 [119]. Indeed, the P250L mutation in 578/10 induced a modification in the structure of the polypeptide that inhibited the formation of NS1 dimers, leading to a decrease in viral replication in vitro on Vero cells [120] as well as neuroinvasiveness in vivo in C57BL/6 mice following intracerebral inoculation.

#### 2.3.2. NS2 Protein

NS2A is a small, multi-functional, hydrophobic protein of 22 kDa involved in RNA replication. NS2A binds to components of the replication complex, to the 3′UTR region of viral RNA, as well as to proteins NS3 and NS5 [121]. NS2A has a role in modulating the host antiviral IFN type I (IFNα/β) response [122,123,124]. NS2B protein (14 kDa) is a co-factor of the NS3 protease with the NS2B and NS3 proteins forming a complex in infected cells [125,126,127].

IFNα/β are essential components of the immune response following viral infection. Control of IFNα/β induction and signaling is therefore essential for virus replication and transmission. Liu and colleagues (2004) investigated the NS2A-A30P mutation in the Kunjin virus. The mutation abolished the capacity of NS2A to inhibit the IFN-β promoter-driven transcription of ISGs (interferon-stimulated genes) during WNV infection [122]. Other studies suggested that other mutations, such as NS2A-D73H and NS2A-M108K [128], or the KUN/NY99 NS2A point mutations A112V, Y119H, M129I, C168R, F212L, and V223I [91], could contribute to NS2A’s reduced capacity to counteract the antiviral response. Further investigations are needed to identify the critical elements in the modulation of the IFN response.

#### 2.3.3. NS3 Protein

NS3 is a multi-domain protein of 70 kDa, with an N-ter protease that constitutes the catalytic domain of the NS2B-NS3 serine protease complex and is known to play a role in cleaving the NS2A/NS2B, NS2B/NS3, NS3/NS4A, and NS4B/NS5 junctions. This N-ter region is also involved in the generation of the C-ter region of the mature capsid and NS4A proteins. NS3 C-ter portion contains the RNA triphosphatase and RNA helicase activities involved in genome capping and viral RNA synthesis and viral replication, respectively [129,130].

Ebel et al. [131] investigated the viral determinants of pathogenesis in mice and viral fitness in mosquitoes and birds, focusing on the RNA helicase domain of the NS3 protein. Because the Asp483 residue is highly conserved among flaviviruses causing encephalitis, such as WNV, TBEV, or JEV, but not in dengue or yellow fever viruses associated with hemorrhagic syndromes [132], it was considered as a possible determinant of neurovirulence and/or neuroinvasion. The mutation of this amino acid was detected in a virus strain attenuated for mice after 20 passages in mosquitoes. The authors reproduced this mutation by reverse genetics and showed that it strongly reduces fitness in vertebrates (chickens), while having little effect on fitness in invertebrate (mosquito) hosts.

Brault et al. [133] have identified a proline within the helicase domain of NS3 at position 249 that is a critical genetic determinant of WNV pathogenesis in American crows. Langevin’s group [134] evaluated the impact of NS3-249 variants on avian and mammalian virulence. They generated viruses presenting multiple amino acid substitutions and inoculated them into two bird species (American crows and house sparrows) and in CD-1 mice. The WNV NS3-249P mutant induced a higher viremia and mortality than the parental WNV NY99 only in the two avian species tested. No difference was seen in mice. This could suggest that this genetic determinant of virulence is specific to avian species. 

Moreover, Sotelo et al. [135] showed that NS3-249 is not the only molecular determinant of virulence for WNV strains bearing glycosylated E. They inoculated different strains of WNV lineage 1 (Spain 2007 (GenBank n° FJ766331), Morocco 2003 (GenBank n°AY701413), and NY99 (GenBank n°FJ151394) into CD-1 mice by the intraperitoneal route. These WNV strains bear either a proline (Spain 2007, NY99) or threonine (Morocco 2003) at position NS3-249. At the dose of 100 PFU, the authors observed 100% mortality regardless of the strain inoculated. At 1 and 10 PFU, however, different clinical outcomes were observed for the three strains. The LD50 was lower for the Morocco 2003 and NY99 strains (1.78 and 2.31 PFU/mL, respectively), while a higher LD50 was observed for WNV Spain 2007 (18 PFU/mL). Consequently, the WNV Morocco 2003 (with a T at NS3-249) was more virulent in mice than WNV Spain 2007 (with P at NS3-249). These results suggest that the amino acid residues at not only position 249 but also at other positions play a role in WNV virulence in mice.

#### 2.3.4. NS4 Protein

NS4A and NS4B are small (16 kDa and 27 kDa) hydrophobic proteins. NS4A plays a role in the virus replication process and may act as a cofactor regulating the ATPase activity of the NS3 helicase [136,137]. NS4B plays a role in immune evasion through inhibition of WNV IFN signaling [138,139] and may also play a role in viral replication [139].

The hydrophobic NS4A protein interacts with cellular membranes through four internal hydrophobic domains. Its N-ter domain is exposed at the cytoplasmic side of ER membranes, such as the NS3 helicase domain. Several studies have demonstrated that NS3-NS4A binding plays a role in polyprotein processing of yellow fever and dengue flaviviruses [140]. Based on these observations, Shiryaev et al. investigated nucleotide changes in the NS4A protein and their impact on NS3 helicase activity by measuring the ATPase activity. The introduction of NS4A-Q46K/Q47K/D50K mutations abolished the ATP-saving functions of NS4A. Taken together, these results suggest that NS4A is essential for optimal performance of the helicase activity of NS3 [136]. 

Several publications have described mutations in the NS4B protein in attenuated and passage-adapted mosquito-borne flaviviruses [141,142,143]. Sequence alignment of multiple flaviviruses showed that these mutations occurred in the same region of the protein. Wicker and colleagues [144] investigated the role of the four cysteine residues (102, 120, 227, and 237) of the NS4B protein using the WNV as a model. They specifically identified the serine–cysteine substitution at position 102 as key to conferring an attenuated phenotype in NIH Swiss mouse models. In vitro, this mutation induced thermosensitivity of the mutant WNV in Vero cells at 41 °C. Other mutations at residues 77 and 125 of the NS4B of DENV serotype 2 were identified as essential to inhibit IFN signaling. It is therefore tempting to consider that the C102S mutation, located in the same region, may play a role in the IFN signaling cascade.

Puig-Basagoiti and colleagues [145] established the adaptive mutation of the Glu to Gly at position 249 (E249G) in the C-ter tail of NS4B as being important for conferring an attenuated phenotype of WNV in mammalian cell cultures and in vivo in mice. In mosquito cells, however, no differences were noted between the parental and mutant virus. Of note, this mutation was also identified in a strain isolated from birds (WNV Texas 2003), which displayed attenuated virulence compared with WNV NY99 [146]. 

In the N-ter region of NS4B, several residues are conserved among the WNV strains (D35, P38, W42, and Y45). Wicker and colleagues demonstrated a role for the proline residue at position 38 in WNV virulence in mammals. The engineered P38G mutant virus was found to contain additional NS4B T116I and NS3 N480H substitutions. The importance of the P38G mutation in the virulence and viral replication was demonstrated in vitro on different cell lines and also in mice [139].

#### 2.3.5. NS5 Protein

The NS5 protein is the largest NS protein (103 kDa) and comprises a N-ter methyltransferase (MTase) and a C-ter RNA-dependent RNA polymerase [147]. By consequence, NS5 is pivotal for flavivirus replication. NS5 catalyzes sequential methylation of a guanine N-7 residue and a ribose 2′O site to generate a type 1 cap on the 5′ end of viral RNA [148]. 

Davis et al. [149] published complementary data on the NS4B-E249G substitution mutation. They demonstrated that the substitution of alanine by valine at position 804 in the C-ter domain of NS5 confers an attenuated phenotype of WNV in cell culture and in vivo in mice. 

Several studies [148,150,151,152,153] have addressed the MTase activity of NS5, and have served to identify three stable attenuating mutations that abolish 2′O methylation activity (K61A, K182A, and E218A). These mutants, however, retain varying degrees of N-7 methyltransferase activity, which seems to be sufficient for virus viability [153]. 

Kaiser and colleagues investigated different single NS5 mutants, including NS5-K61A and NS5-E218A, as well as the double mutant NS5-K61A/E218A, in the framework of a NY99 IC. Although each single mutant was attenuated for neuroinvasiveness in NIH Swiss Webster mice, the double mutant was not, though the mice survived for a longer period than those challenged with the parental NY99 strain. This unexpected result was due to the reversion of the double mutant at both mutation sites. To demonstrate the reversion of this double mutant, Kaiser et al. realized next-generation sequencing on the double mutant viral stock after many passages on Vero cells. At P0, 3.3% of viral stock presented the A61K reversion and 2.8% the A218E reversion. After one passage, the A61K reversion was detected in 5% of the viral population and A218E in 4.1%. Finally, at passage 5 on Vero cells, the A61K and A218E reversions made up 41% and 47.6%, respectively, of the population. Nonetheless, the NS5-K61A and NS5-E218A mutations in the N-ter methyltransferase domain of NS5 are important virulence determinants of the WNV and appear to be promising targets to generate candidate live WNV vaccines [154].

### 2.4. The 3′UTR Non-Coding Region

The 5′ and 3′ UTR flanking the single ORF of flaviviruses act as important regulators of viral genome replication and translation. They contain highly elaborated secondary structures [155]. The 3′UTR of flaviviruses ranges between 380 to 600 nt in length and can be divided into three domains: a highly variable proximal domain 1 that follows the stop codon, a second domain 2 with a moderately conserved sequence and a number of stem-loops and dumbbell structures, and the highly conserved distal domain 3 [156]. 

In addition to the full-length genomic RNA (gRNA), an RNA sequence of approximatively 0.5 kb—called subgenomic flaviviral RNA (sfRNA)—has been detected in flavivirus-infected cells [157,158]. Recent studies have shown that sfRNA is a degradation product generated by a host enzyme, probably the 5′-3′ exoribonuclease XRN1 [158,159]. It is thought that complete degradation of gRNA by XRN1 may be prevented by the 3′UTR secondary structure. The generation of sfRNA plays a role in WNV virulence [160]. A study conducted by Roby and colleagues [156] demonstrated that deletion mutants of WNV incapable of producing sfRNA were attenuated both in cell culture and in vivo in mice. 

Davis and colleagues [149] sought mutations in the 3′ UTR involved in WNV virulence. They determined that the following substitutions—3′UTR-A10596G/C10774U/A10799G—did not by themselves change the WNV phenotype. However, WNV virulence was attenuated when these three mutations occurred simultaneously with the NS4B-E249G mutation.

## 3. Avian Model

In many bird species, the level of viremia is sufficient to infect mosquitoes. They are not only reservoirs of WNV but also amplifiers of the virus and a source of infection for dead-end-hosts. Severe WNV disease has been diagnosed in Accipitriformes and Passeriformes, and among the latter, especially in Corvidae. Clinical signs in susceptible species include ruffled feathers, lethargy, ataxia, inability to fly, seizures, and unusual postures. 

A variety of wild bird species (at least 77 species belonging to 29 families and 12 orders) have been experimentally inoculated with different WNVs to estimate their host competence and to study WNV pathogenesis and virulence [49]. Passeriformes and Charadriiformes are considered to be highly competent hosts [48]. The virulence determinants of WNV have mainly been characterized in the American crow (AMCR) (*Corvus brachyrhynchos*), house sparrow (HOSP) (*Passer domesticus*), red-legged partridge (*Alectoris rufa*) [49], house finch (HOFI) (*Haemorhous mexicanus*), and young specific-pathogen-free (SPF) chicks (*Gallus gallus domesticus*) [161,162]. Host competence should be considered when evaluating WNV fitness and virulence in birds [49]. Bird species with low (chicken), moderate, and high (AMCRs and HOSPs) competence will differentially replicate the WNV and, consequently, evaluation of virulence factors in vivo will be influenced to some extent by the bird species used. The geographic distribution and abundance of avian species will also influence the amplification and transmission dynamics locally. Only a few of the identified molecular determinants of virulence identified in mice have been confirmed in natural avian hosts (Table 1 and Figure 7).

### 3.1. Structural Proteins

#### 3.1.1. E Protein

Several studies [76,163] have confirmed the role of glycosylation of the E protein in viral pathogenicity in birds, both in vitro (on avian cells) and in vivo (on two-day old chicken). Brault et al. [71] confirmed that Mexican variants of WNV with a glycosylated E motif (E-P156S) produced a higher viremia and shorter survival time in American crows and house sparrows, thereby defining the glycosylation state of E protein as a virulence determinant in birds. However, strains that are non-neuroinvasive in mice cause significant mortality in birds, suggesting that mechanisms of virulence and attenuation may vary between vertebrate hosts. Therefore, results acquired in mouse models may not accurately predict virulence in birds. Moreover, among avian species, results obtained in a given species cannot be transported to other susceptible species and need to be assessed experimentally.

#### 3.1.2. prM Protein

In comparison with the highly virulent NY99 strain, the strains isolated in Mexico, Central America, or South America in 2003 were attenuated [164,165,166]. A study conducted by Brault and colleagues in 2011, found that the WNV Mexican variants, whether the E protein was glycosylated or not, were less virulent than the NY99 strain, suggesting that other determinants impact the virulence in the avian model. Langevin et al. generated chimeric mutants between the Mexican and the NY99 strains and inoculated them into three bird species (AMCRs, HOSPs, and HOFIs). They demonstrated that the prM-I141T and E-S156P mutations decrease WNV NY99 virulence and that the reciprocal mutations prM-T141I and E-P156S raise the virulence of the Mexican strain in birds [162].

### 3.2. Non-Structural Proteins

#### 3.2.1. NS1, NS2, and NS4 Proteins

Andrade et al. [167] evaluated temperature sensitivity using WNV NY99 and COAV997 (WNV California 2003) strains that differed at five non-synonymous mutations. They infected duck embryonic fibroblasts (DEF) cells with parental and chimeric viruses at three temperatures, 37 °C, 41 °C, and 44 °C, corresponding to temperatures observed in birds that are slightly, moderately, and severely ill, respectively [168]. They observed that the NS1-K110N and the NS4A-F92L substitutions in WNV NY99 decreased cell growth only at 44 °C. These results highlight the importance of residues 110 (asparagine) of the NS1 protein and 92 (phenylalanine) of the NS4A protein in WNV infection of avian cells. A similar test was carried out in vitro on *Aedes albopictus* mosquito cells (C6/36) by studying infection at 22 °C, 28 °C, and 34 °C, corresponding to temperatures typical of spring, early summer, and summer, respectively. No differences were observed in C6/36 cells, suggesting that certain genomic regions could be involved in temperature adaptation in the host but not in the vector.

The NY99 strain has been extensively studied and is widely used as a model strain for WNV studies. An alternative lineage 1 WNV strain isolated in Kenya (WNV KN3829) shows only eleven amino-acid differences with the NY99 strain. The two strains exhibit a different virulence phenotype in American crows. Although NS3 studies show the importance of the NS3-249T substitution in attenuation of WNV virulence in birds, viremia was lower for the KN3829 parental strain and KN3829-T249P mutant than the NY99-P249T and NY99 strains, respectively, suggesting the importance of other mutations for the avian virulence phenotype. Dietrich et al. [169] focused on the role of NS1 and NS2 proteins in addition to the NS3-P249T substitution in bird virulence. In an in vivo study, viremia was shown to be higher for mutants having either a proline or a threonine residue at position 249 in the NS3 protein along with the NS1 and NS2B regions of NY99, than for the parental viruses. These regions differ by three residues between the two strains (residues 70 of NS1; 52 and 103 of NS2). Site-directed mutagenesis was then used to replace low virulence strain residues with those of a high virulence strain (NS1-S70A, NS2A-A52T, and NS2B-A103V). Significant differences were not observed in vivo, whether for viremia or mortality, suggesting that the difference in virulence does not depend on a single residue. They also evaluated the temperature sensitivity of WNV NY99, KN3829, and the chimeric mutants thereof. They infected peripheral blood mononuclear cells (PBMC) of AMCR and DEF at different temperatures. Replacement of the NS1-NS2B region of KN3829 by the corresponding fragment of NY99 increased the viral titer in DEF.

These results show that differences in the NS1 and NS2B region may influence the pathogenicity of WNV in bird models, independently of the NS3-249 substitution. Moreover, it seems that modification of the NS1 and NS2B proteins impacts the strain-specific temperature sensitivity. However, the temperature of viral replication directly influences the capacity of the WNV to infect vectors and hosts. Indeed, to increase the probability of survival, WNV has to be able to replicate at temperatures ranging from 14 °C (minimal temperature for mosquito life cycle) to 45 °C (maximal temperature of febrile bird). It is frequently observed that a poor in vitro adaptation of viral strains to different temperatures is correlated with an attenuated virulence phenotype in vivo [170,171,172].

#### 3.2.2. NS3 Protein

The NS3-T249P is present in many WNV strains that have caused major outbreaks in humans, such as in Egypt (1950), Romania (1996), Russia (1996), New York (1999), and Israel (1997–1998) [134].

Brault and colleagues also identified the NS3-249P residue as a critical determinant of WNV virulence in American crows. To this end, they compared the viremia of AMCRs infected with WNV NY99 (highly virulent), KN3829 (attenuated strain), and the mutants NY99-P249T and KN3829-T249P [133,173]. The attenuated strain KN3829 and the mutant NY99-P249T gave rise to a low level, delayed viremia at day 3 pi, compared with high titers obtained for NY99 and KN3829-T249P. However, studies conducted by Langevin et al. in 2005 and Sotelo et al. in 2011 in other avian species, HOSPs and red-legged partridges, respectively, did not validate these results. This observation suggests that results obtained in a given species cannot be transported to other susceptible avian species and need to be validated experimentally. 

Langevin and colleagues demonstrated that, depending on the nature of the substitution at position 249 of NS3 (tested substitutions: NS3-P249A, NS3-P249D, NS3-P249H, and NS3-P249T), in vivo virulence in AMCRs and HOSPs is differentially affected. No differences in viremia was observed in AMCRs infected with NS3-P249D and NS3-P249H mutants or the parental strain NS3-249P. The NS3-P249T mutation, however, drastically reduced the viremia and diminished the lethality. Modification of the NS3-249 residue had an impact on viremia in HOSPs. Unlike observations made in AMCRs, the NS3-249T mutant produced a higher viral titer than the NS3-249P mutant, but rapidly declined at 3 dpi. These results confirm the importance of the NS3-249 residue in avian virulence, especially in the Israelo-American clade, which is the only group of lineage 1 WNV strains with high pathogenicity in birds [134]. Dridi’s group [73] also investigated the NS3-249P residue. They performed an in vivo experiment in young SPF chickens, inoculated with three different lineage 2 WNV strains: Hun2004 and Aus 2008, which have a NS3-249H residue, and Gr2011, which has a NS3-249P residue. After subcutaneous inoculation, viremia peaked for all viruses at 2 dpi. However, for Gr2011, an infectious virus could only be detected at this time point. Moreover, the infectious titer was significantly higher for Hun2004 and Aus2008 than for Gr2011. These results show that, despite its NS3-249P residue, Gr2011 is less virulent than the other strains tested. That could suggest that NS3-249P is neither sufficient nor necessary for WNV virulence in birds.

Viral fitness was evaluated by Ebel et al. [131] in mosquitos and chicken. Competitive inoculation of both wild-type WNV and mutant NS3Δ483 showed that the NS3Δ483 mutant was outcompeted by wt WNV in chickens, suggesting a role for NS3 helicase activity in WNV virulence.

## 4. Insect Model

Using the mosquito as an in vivo model can be complex depending on the route used to inoculate the virus. Experimental techniques for WNV infection in mosquitoes vary from thoracic microinjection and membrane blood feeding to blood feeding directly on mice [174]. The origin of the blood source is also an important criterion in the study of vector competence of mosquitoes. Avian blood, especially chicken blood, is ideal [68]. Experimental infections with WNV in mosquito vectors have mainly been performed to assess the replication efficacy in mosquito hosts and therefore deduce virus circulation in the natural enzootic transmission cycle. Studies of WNV virulence have first addressed multiplication and dissemination of WNV variants in vitro in cell lines, essentially on C6/36 (*Ae. albopictus* cells), and then sought confirmation in vivo, mainly in *Cx. pipiens* mosquitoes. Other studies have been performed on *Ae. albopictus*, *Ae. caspius*, *Ae. detritus*, and *Cx. modestus* [68].

### 4.1. Structural Proteins

#### E Protein

Murata et al. examined the role of glycosylation of the E protein in viral dissemination and replication for two mutant viruses, 6-LP (S156, glyE+, GenBank No. AB185914) and 6-SP (P156, glyE-, GenBank No. AB185915) in C6/36 cells and in vivo in *Cx. pipiens*. In vitro temperature sensitivity tests and in vivo experiments revealed that the mutations had no effect on multiplication and dissemination of WNV [64]. Therefore, glycosylation of the E protein is not critical for the virus to propagate and disseminate within the vector, at least when the virus is inoculated intrathoracically into mosquito hemolymph so as to bypass the midgut barrier [110].

Another study conducted by Moudy et al. [175] focused on the N154I substitution within the E protein of WNV NY99. No differences were observed between the parental and mutant virus. These results do not agree with conclusions from previous studies [92,176], which demonstrated a 10-fold decrease in WNV titers in mosquito cells. This difference could possibly be explained by the strain of WNV used. In vivo experiments were performed on *Culex pipiens* and *Culex tarsalis*. The authors tested two inoculation routes (perioral and intrathoracic) to infect female mosquitoes, in which the inoculation route, replication of the NY99 N154I variant, was lower than that of the parental virus in the two mosquito species tested. This result highlights the importance of the asparagine residue at position 154 of the E protein in WNV replication in mosquitoes. The currently known genetic determinants of virulence in mosquitoes are represented in Figure 8.

### 4.2. Non-Structural Proteins

#### 4.2.1. NS1 and E Protein Glycosylation

Van Slyke and colleagues [177] carried out in vivo experiments to study vector competence of *Culex tarsalis*. They investigated three mutations that remove NS1 glycosylation (NS1-130-132QQA/175A/207A) alone or in addition to ablation of the glycosylation site of the E protein (E154S-NS1-130A/175A/207A). Both mutants were attenuated for vector competence in *Culex tarsalis*. Infection, dissemination, and transmission were more impaired for the NS1-130-132QQA/175A/207A mutant than for the mutant E154S-NS1-130A/175A/207A. The NS1 glycosylation is therefore an important determinant for WNV dissemination and transmission in *Culex tarsalis*.

#### 4.2.2. NS3 Protein 

Ebel et al. [131] evaluated the fitness of the WNV NS3Δ483 mutant in *Culex quinquefasciatus.* A lower to moderate fitness was observed for the mutant WNV compared with that of the parental counterpart. The authors also demonstrated reduced fitness for the WNV NS3Δ483 mutant in chicken. The RNA helicase of NS3 thus plays a role in fitness in both avian and mosquito hosts.

#### 4.2.3. NS4 Protein 

Van Slyke et al. [177] studied the impact of several mutations in NS4B (NS4B-P38G/T116A, NS4B-C102S, NS4B-P38A, and NS4B-E249G) on the vector competence of *Culex tarsalis*. They found that all the NS4B mutations increased the vector competence of *Culex tarsalis*, as evidenced by a higher infection rate, dissemination, and transmission for these viruses. These results are discordant with observations in mammals in which the corresponding NS4B mutants had attenuated phenotypes in neurovirulence [144].

## 5. Discussion

WNV is expanding across Europe and other continents. It re-emerged in Europe during the summers of 2016–2018, with an increase in reported autochthonous cases (ECDC, update 9 November 2018). This flavivirus is maintained in an enzootic cycle involving different species of birds and mosquitoes with humans and horses identified as dead-end hosts [47,48,178]. It is important to acquire better clarification for several aspects of WNV circulation, including genomic diversity, pathogenicity, and transmissibility of the WNV strain. Many studies have provided insight into the profile of circulating WNV isolates. Some isolates from birds and mosquitoes are naturally attenuated [146,179]. For instance, strains isolated in Texas (Houston) in 2003 are temperature sensitive and attenuated in mice and birds. Genomic sequence analyses did not identify specific mutations related to this phenotype. Authors have described the NS4B E249G substitution and additional mutations such as prM-N4D and NS5 A804V, all or most of which are present in all but a few of these attenuated isolates. Therefore, natural attenuation probably relies on the accumulation of multiple specific mutations in different parts of the genome. Thanks to reverse genetic technologies and site-directed mutagenesis, research groups have successfully investigated the impact of individual mutations engineered in the structural, prM/M, and E genes, and non-structural NS1, NS2A, NS3, NS4A, NS4B, and NS5 genes. 

WNV is biologically diverse, with up to eight different lineages proposed. The majority of the studies performed to identify the critical molecular determinants of virulence have used WNV NY99 as a reference strain. It would be therefore of great interest to look for molecular determinants of virulence in WNV strains isolated more recently in Africa, Europe, and Australia. The identification of virulence determinants is a key step in understanding WNV pathogenesis and in designing preventive and therapeutic tools. The study of genetic expression, replication, and pathogenesis of flavivirus, especially WNV, has been facilitated by the improvement of reverse genetic technologies. In order to overcome the toxicity of full-length cDNA in bacteria, various approaches have been employed, such as long PCR, CPEC, ISA, and ISA-derived methods [83]. These methods allow rapid generation of infectious flavivirus cDNA. The production of such tools was essential in acquiring the current high level of knowledge about flavivirus virulence determinants. Most studies were initially undertaken in mouse models but rapidly the research community moved on to assess WNV virulence in more relevant models, especially birds, the natural hosts of the virus, and its vectors, especially *Culex* spp. mosquitoes. A review edited by Kaiser and colleagues [88] summarized the key molecular determinants implicated in WNV virulence in mammals and mosquitoes. Four years later, the present review updates the current understanding of the virulence determinants of WNV in mammals, birds, and mosquitoes, and provides a comparison of the differences observed between the three models and with other flaviviruses (Table 2). Several groups have compared the role of critical virulence determinants in the three models. Such is the case of Murata and colleagues [76], who focused on the glycosylation state of the E protein. In vitro experiments performed in mammalian, avian, and mosquito cell lines demonstrated that the WNV gly+E protein is temperature-sensitive in vertebrate cells only. These results were validated in vivo. Glycosylation of the E protein increased the virulence in avian and mammalian hosts but had no impact on the replication and dissemination in *Culex pipiens*. Another group [131] focused on the NS3 protein, especially NS3Δ483. This deletion is conserved in other flaviviruses, causing encephalitis such as TBEV and JEV, but not in dengue and yellow fever virus [132]. They observed an attenuated phenotype in mice, and a decreased fitness of WNV in birds and mosquitoes. These two studies highlighted the necessity to perform comparative studies in the natural hosts and reservoirs of WNV to appreciate the global impact of mutations in WNV pathogenesis. In addition, the dynamics and stability of virulent/non virulent mutations through serial cross-passages in mammalian/avian/arthropod hosts would deserve further studies. Researchers now have a robust inventory of specific virulence determinants across most of the WNV genome underpinned by phenotypic studies in vertebrate and/or invertebrate hosts/reservoirs. The identification of key mutations accounting for WNV attenuation and pathogenesis need to be pursued. The proper characterization of virulence determinants requires an extensive analysis of the modifications induced in the complete gene-product and its effect on neuroinvasiveness, tissue tropism by immunohistochemistry, and immunopathogenesis. It would be interesting to start investigating the impact of combined mutations in several genes. Indeed, according to the phenotypic studies conducted on WNV isolates, the level of virulence depends upon multiple sites in the genome. This would help in elucidating the pathobiological mechanisms of flavivirus infection that underline the virulence determinants. This approach, combined with reverse genetic and site-directed mutagenesis, would be useful for design of a safer WNV vaccine containing the mutations of interest.

It would also be a good starting point to compare the virulence and pathogenesis of the WNV with those of other flaviviruses of interest, such as the dengue and yellow fever viruses [97]. It is noteworthy that the study of flavivirus molecular determinants has been driven by the identification in the field or by artificial selection of mutants with properties of interest, such as attenuation, neuroadaptation, and escape from neutralization. Much work [207] has been performed on *Culex*-borne JEV and WNV, *Aedes*-borne YFV and DENV, and tick-borne TBEV, while more limited datasets are available for other or recently emerging flaviviruses (ZIKV, for example). As a consequence, most of the molecular determinants shown to influence WNV, JEV, TBEV, MVEV, YFV, DENV, or ZIKV virulence or vector-borne transmission have only been studied for a single flavivirus [208,209]. Moreover, differences in the topology of NS proteins between flaviviruses, such as differences in NS2A aa residues exposed at the luminal side of the endoplasmic reticulum or in the cytoplasm of YFV and DENV infected cells, might explain, in part, the apparent variation in critical residues identified for different flaviviruses [210]. Of note, a comparison of the molecular determinants of flaviviruses has identified a number of these that are conserved in most flaviviruses, independent of the nature of the vector (mosquitoes vs. ticks, with TBEV) or of their tropism (neurotropism for JEV, WNV, and TBEV vs. viscerotropism for YFV and DENV (Table 2)); these shared determinants are likely to correspond to protein domains or functions that are critical for flavivirus biology, such as N-glycosylation domains in prM, E, and NS1, as well as E receptor-binding and fusion domains. Only one residue, E-138, has been shown to modulate the neurovirulence of JEV and WNV in a differential manner, as the E-E138K mutation diminished the neuropathogenicity of JEV in mice but had no impact on WNV [113,186,189] (Table 2). The identification of attenuating non-synonymous mutations that are conserved among flaviviruses could help in the rational design of attenuated vaccines against emerging flaviviruses, such as the Usutu or Zika viruses. 

Finally, we actually face environmental changes (climate change, ecosystem changes due to biodiversity loss, and land-use modification). In response to global warming, bird species, some of them competent for WNV, may have to change their distribution area, possibly spreading WNV into new regions. In addition to altering the distribution of wildlife hosts, global warming will impact the mosquito distribution globally. Global warming will lead to the desertification of many regions that will become unfavorable to mosquitoes, but at the same time it could make temperate and cold regions more favorable to mosquitoes. In this context, WNV distribution may change in the coming years, leading to recurrent outbreaks in hitherto spared areas. We know that a fast reaction to a new outbreak is one of the best ways to control an epidemic. Thanks to modern molecular biology, viral genomes can be quickly sequenced. Detailed knowledge of the molecular determinants of virulence present in the viral genome would certainly help to anticipate the possible grade of WNV epidemics and, hence, properly implement control measures. 

## Figures and Tables

**Figure 1 ijms-21-09117-f001:**
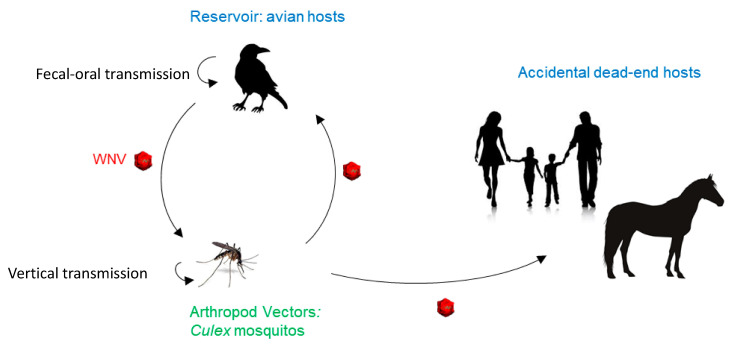
West Nile virus (WNV) transmission cycle. In nature, WNV transmission occurs through an enzootic cycle between birds and mosquitoes, the latter mainly of the genus *Culex*. Humans and mammals can be infected but are considered to be dead-end hosts.

**Figure 2 ijms-21-09117-f002:**
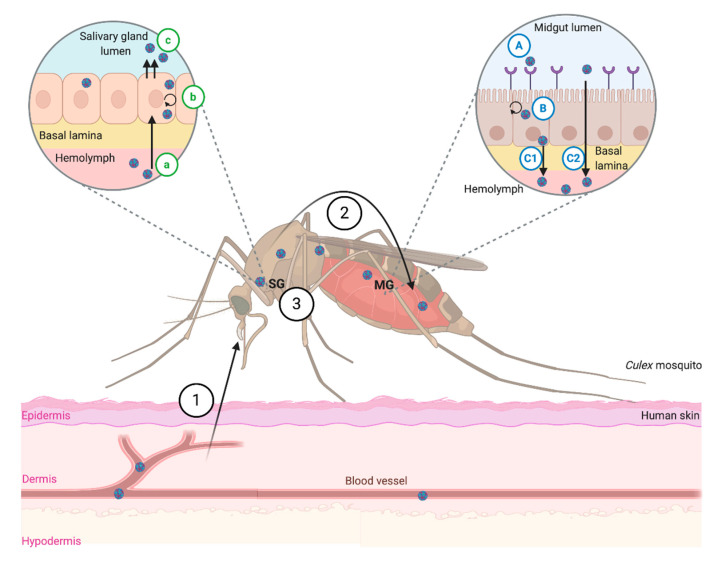
Dissemination steps of WNV in the invertebrate vector. (**1**) Mosquito feeds on virus-infected blood. (**2**) WNV infects mosquito midgut: (**A**)—Virus binds to epithelial cells by a protein receptor. (**B**)—Viral replication in midgut cells. (**C1**)—Direct passage from basal lamina to hemolymph (1st way). (**C2**)—Direct paracellular passage through midgut cells (2nd way). (**3**) Dissemination of WNV to other organs like salivary glands: (**a**)—Infection of epithelial cells by direct passage from hemolymph to cells. (**b**)—Viral replication in epithelial cells. (**c**)—Release of virus from cells to salivary gland lumen, with or without apoptosis of epithelial cells (inspired by [68,69]).

**Figure 3 ijms-21-09117-f003:**
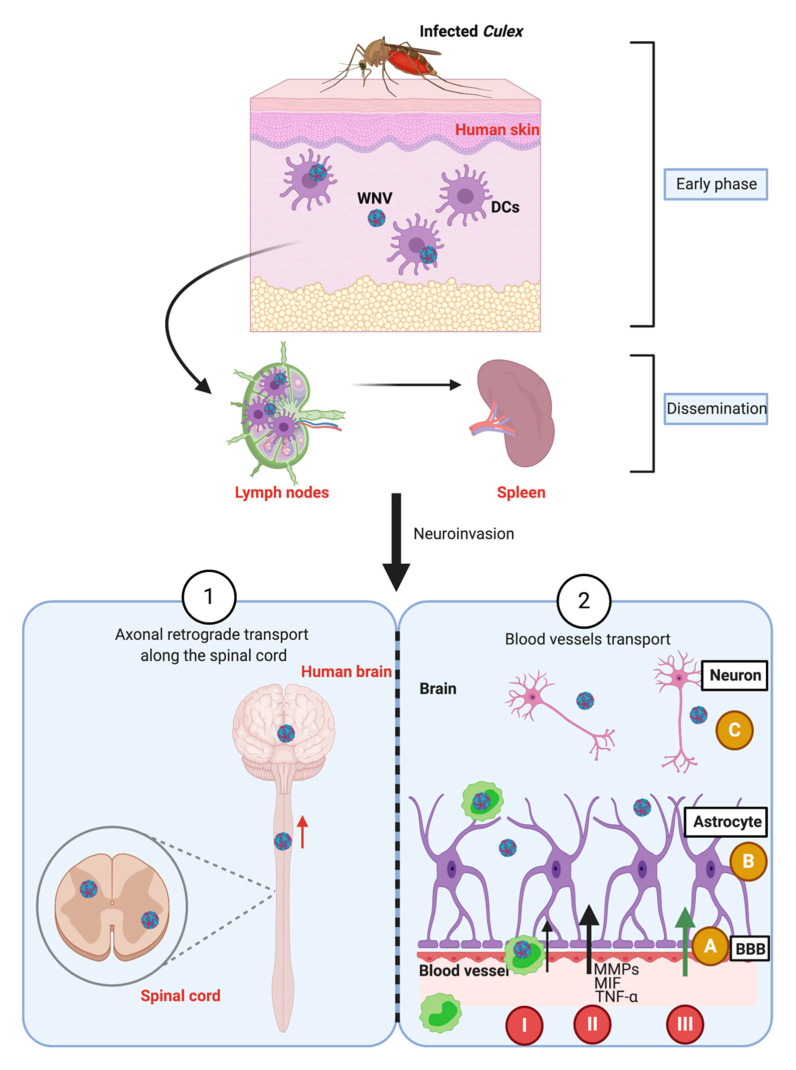
Dissemination steps of WNV in vertebrate hosts such as humans. At the early phase of infection, WNV disseminates in the human skin after a WNV-infected mosquito bite. First, replication begins in the keratinocytes and dendritic cells (DCs) and is followed by a migration of the WNV into the dermis. After that, DCs containing the WNV migrate to the lymph nodes where the WNV is amplified. Systemic viremia gives rise to infection of peripheral organ infection, such as the spleen. More severe infections lead to neuroinvasion by two possible routes. The first (**1**) consists of an axonal retrograde transport along the spinal cord [70]. The second (**2**) involves blood vessel transport and crossing over the blood–brain barrier (BBB). Three steps are required to mediate the neurovirulence in the brain by the blood route: (**A**)—Crossing the BBB. (**B**)—The virus interacts with the brain cells. (**C**)—Infection of neurons. To cross the BBB, three methods are possible. (**I**)—The first is a “Trojan horse” mechanism, in which the WNV is internalized into lymphocytes that are able to cross the BBB. (**II**)—The second method is permeabilization of the BBB in response to TNF-α and MIF secretion. After MIF secretion by leucocytes, the MMPs are produced and increase the BBB permeability [51,52,53,54]. (**III**)—The third is enhancement of virus attachment, probably dependent on E protein glycosylation, which permits transcellular passage (inspired by [11,65]).

**Figure 4 ijms-21-09117-f004:**
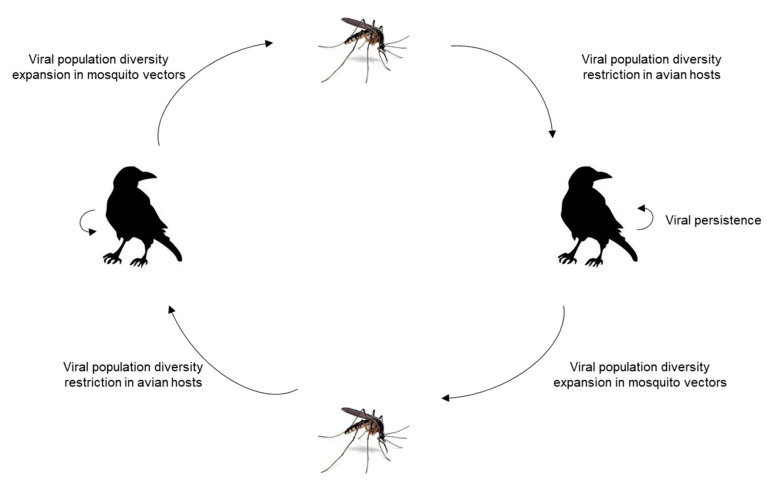
WNV evolution through genetic diversity balance between the avian hosts and mosquito vectors. Viral diversity increases in mosquitoes and decreases in birds.

**Figure 5 ijms-21-09117-f005:**
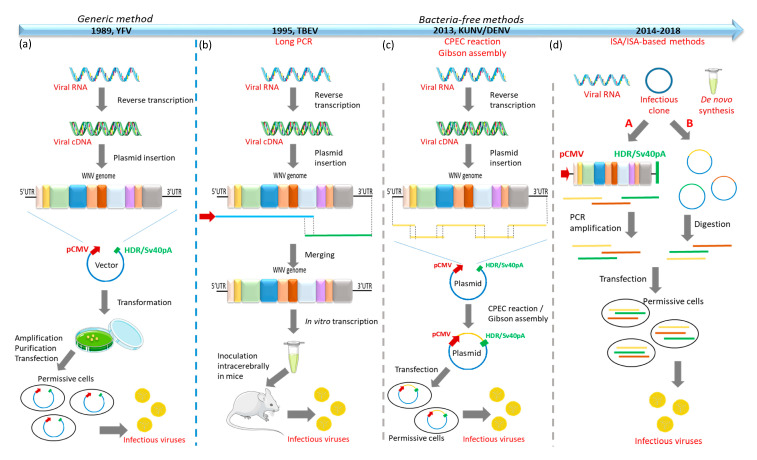
Overview of the main reverse genetic methods. (**a**) First generic method for reverse genetics developed in 1989 for YFV. cDNA was produced from viral single-stranded RNA, and then inserted into a plasmid upstream of a pCMV promoter and downstream of the hepatitis delta ribozyme, followed by the simian virus 40 polyadenylation signal (HDR/SV40(pA)) sequence. After amplification and purification, the construction was introduced into permissive cells by transfection, and infectious clones were obtained. (**b**) Bacteria-free method used for TBEV in 1995. Two long PCR products were synthetized and joined by using restriction enzymes or fusion PCR. After in vitro transcription, infectious clones were generated after intracerebral inoculation of mice with the long PCR fragment. (**c**) Bacteria-free method using the CPEC (KUN) or Gibson reaction (DENV), 2013. Multiple PCR amplicons were inserted into a plasmid containing pCMV and HDR/Sv40(pA) sequences and joined by CPEC or Gibson reaction. Infectious clones were obtained after transfection of permissive cells. (**d**) New bacteria-free method for reverse genetics in flaviviruses. (**A**)—In the infectious subgenomic amplicons (ISA) method the genetic material consists of viral RNA, infectious clones, or de novo synthesis. PCR products that cover the entire genome are used for direct transfection of permissive cells. (**B**)—The new ISA-based reverse genetic method, called the subgenomic plasmids recombination method (SuPReMe), resembles the ISA method, but the genome fragments are cloned into plasmids at the restriction sites. After restriction enzyme digestion, genomic fragments are used for transfection of permissive cells, and infectious clones are produced (according to [79]).

**Figure 6 ijms-21-09117-f006:**
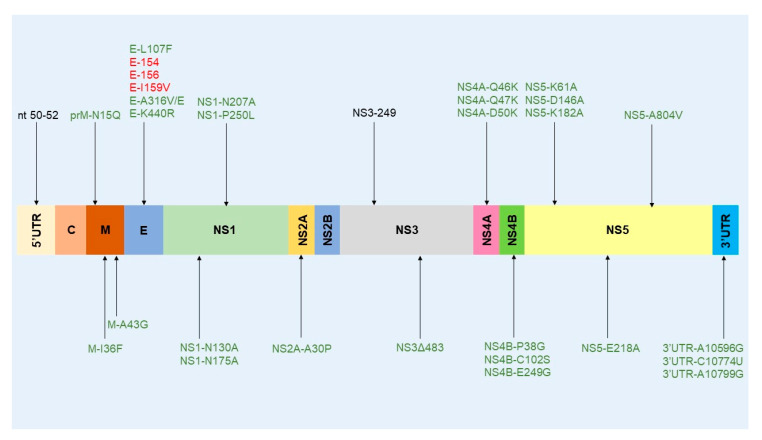
Virulence determinants in the WNV genome found in mammalian hosts. Molecular virulence determinants found in vitro or in vivo in mammalian cells (Vero, A549) and mouse models are represented throughout the entire WNV genome. Green: mutations involved in the attenuated viral phenotype. Red: mutations involved in the more virulent viral phenotype (adapted from [88]).

**Figure 7 ijms-21-09117-f007:**
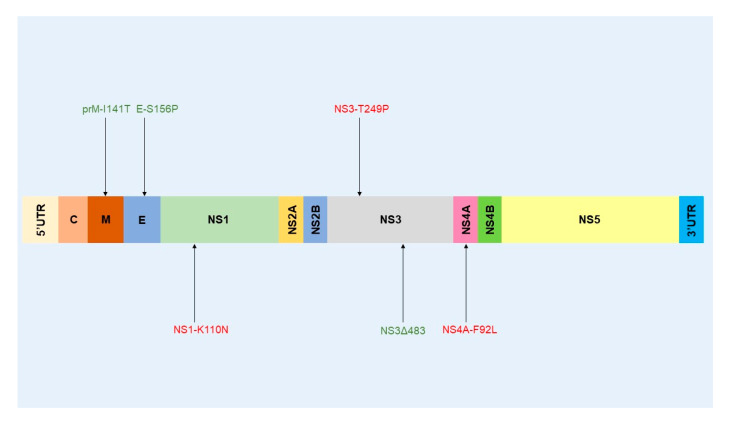
Virulence determinants in the WNV genome found in avian hosts. Molecular virulence determinants validated in vitro in the DEF avian cell line or in vivo in American crows, house sparrows, red-legged partridges, and specific-pathogen-free (SPF) chickens. Green: mutations involved in attenuated viral phenotype. Red: mutations involved in more virulent viral phenotype.

**Figure 8 ijms-21-09117-f008:**
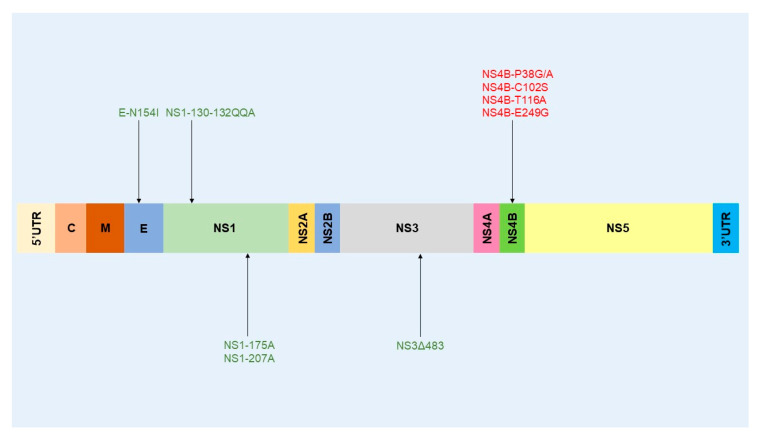
Virulence determinants in the WNV genome found in mosquito vectors. Molecular virulence determinants validated in vitro in the C6/36 mosquito cell line or in vivo in *Culex pipiens* and *Culex tarsalis*. Green: mutations involved in attenuated viral phenotype. Red: mutations involved in more virulent viral phenotype.

**Table 1 ijms-21-09117-t001:** Comparison of molecular virulence determinants of WNV in mammalian, avian, and insect models. The table details most of the residues involved in virulence variation in vivo and in vitro, as well as mutations or deletions most often observed. Green: mutations that decrease virulence. Red: mutation that increase virulence.

	Mammals Model		Avian Model		Insect Model	
	nucleotides involved/mutation/deletion	effect	nucleotides involved/mutation/deletion	effect	nucleotides involved/mutation/deletion	effect
5′UTR	5′UTR (nt 50-52)					
prM/M	prM-N15Q	decreases viral RNA quantity	prM-I141T	decreases virulence		
	M-I36F	small plaque phen., att phen.				
	M-A43G	small plaque phen., att phen.				
E	glyE+ (nt 154-156)	increases neuroinv., neurovir.	E-S156P (glyE-)	decreases viremia	E-N154I (glyE-)	decreases viral repli.
	E-L107F					
	E-159					
	E-A316V/E	decreases virulence				
	E-K440R	decreases virulence				
NS1	NS1-N130A	stop neuroinv.	NS1-K110N	enhance viral temp. resistance	NS1-130-132QQA	decreases infection, dissemin., transm.
	NS1-N175A	stop neuroinv. & neurovir.			NS1-175A	decreases infection, dissemin., transm.
	NS1-N207A	stop neuroinv.			NS1-207A	decreases infection, dissemin., transm.
	NS1-P250L	decreases viral titer and stop neuroinv.				
NS2	NS2A-A30P	decreases viral repli., att. for neuroinv. and neurovir.				
NS3	NS3△483	decreases mortality	NS3△483	decreases virulence	NS3△483	decreases fitness
	NS3-249		NS3-T249P	decreases virulence		
NS4	NS4A-E46K		NS4A-F92L	enhance viral temp. resistance		
	NS4A-E47K				NS4A-T116A	
	NS4A-D50K					
	NS4B-C102S	temp. sens., decreases neuroinv. and neurovir.			NS4B-C102S	better mosquito transm.
	NS4B-E249G	decreases viral repli. and mortality			NS4B-E249G	
	NS4B-P38G	temp. sens., small plaque, decreases mortality			NS4A-P38G/A	better mosquito transm.
NS5	NS5-A804V	att. for neuroinv.				
	NS5-K61A	decreases repli., no lethality				
	NS5-K182A	decreases repli.				
	NS5-E218A	decreases repli., no lethality				
	NS5-D146A	decreases repli.				
3′UTR	3′UTR-A10596G	decreases virulence				
	3′UTR-C10774U	decreases virulence				
	3′UTRA10799G	decreases virulence				

**Table 2 ijms-21-09117-t002:** Comparison of molecular virulence determinants among flaviviruses. The table details residues involved in virulence variation in vivo and in vitro and shared by at least two flaviviruses. TBEV: tick-borne encephalitis virus; DENV: dengue virus; ZIKV: zika virus; YFV yellow fever virus; JEV: Japanese encephalitis virus; MVEV: Murray valley encephalitis virus [92,97,99,102,106,107,108,112,113,116,177,180,181,182,183,184,185,186,187,188,189,190,191,192,193,194,195,196,197,198,199,200,201,202,203,204,205,206].

	Location	Virus(es)	Residues	Similar Residues in WNV Genome	Effects	References
prM/M	prM glycosylation sites	TBEV	D143, R144	N15	ablation of prM glycosylation sites impacts virus assembly and infectivity and enhances TBEV neurovirulence	[92,180]
	ApoptoM	DENV, YFV	L36	I36	modulates the death-promoting activity of M, virus replication and neurovirulence	[97,99,181]
E	Domain I (N-glycosylation site)	TBEV	N154	154–157	modulates virus secretion from infected cells and virus infectivity in mammalian cells but not in arthropod cells, as well as virus replication and neuroinvasiveness in in vivo models	[182]
		ZIKV	T156			[183]
		ZIKV	154–157			[102,106,107,108,184]
	Hinge region linking Domains I and II	JEV, DENV, YFV	E49 (JEV), Q52	not investigated on WNV	impairs endocytosis and modulates neuroinvasiveness and neurovirulence	[185,186,187,188]
		JEV	E138K	E138	reduces virus replication, neuroinvasion and neurovirulence for JEV, neurovirulence not affected for WNV and residue prone to rapid reversion in WNV	[113,186,189]
	Domain II (fusion peptide)	JEV, YFV	L107F	L107	impairs fusion, decreases viral growth in mammalian and insect cells and neuroinvasiveness in mice but does not affect neurovirulence	[112,116,187,190]
		DENV	G102S, F108A			[191]
	Domain III (receptor binding site)	JEV	E306K	A316	influences binding to glycosaminoglycans (residues 325-326 and 380) or other cell receptors, modulates the efficacy of virus spread, neuroinvasiveness and neurovirulence - diminishes infection rates in Aedes aegypti mosquitoes (YFV), not studied in Culex mosquitoes	[192]
		YFV, DENV	S305F			[181,193,194,195]
		TBEV	D308K			[197]
		JEV	A315V			[198]
		YFV	S325P, E326K/R			[193,195,196]
		YFV, MVEV	R380T			[193,194,199,200]
NS1	N-glycosylation site	DENV	N130A + N208A	N130 + N207	ablation of the first glycosylation site (N130) decreases replication, viral production and neurovirulence and diminishes vector competence of Culex tarsalis mosquitoes for WNV	[177,198]
		YFV	N130A + N208A		Decreased replication and neurovirulence (1st glycosylation site)	[116,201,202]
3′UTR	Deletion of nucleotides	JEV	−27nt	not investigated on WNV	Attenuates or increases (TBEV) neurovirulence	[203]
		DENV	−4 nt			[204,205]
		TBEV	−206 nt			[206]

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
