# Peer review of "Molecular Determinants of West Nile Virus Virulence and Pathogenesis in Vertebrate and Invertebrate Hosts"

_ijms, 2020, doi:10.3390/ijms21239117_

Round 1

Reviewer 1 Report

 Summary

The review manuscript “Molecular determinants of West Nile virus virulence and pathogenesis in vertebrate and invertebrate hosts” by Fiacre and colleagues is well written with an appropriate balance between in-text data/information and graphical presentation. The individual sections are comprehensive, yet succinctly summarized and the body of literature cited is adequate.

Major comments

  • Please double check the entire text for proper use of grammar. The majority of the text is written in good English, but there are certain paragraphs (the abstract is one of them) that would benefit from a grammar check.
  • The concluding paragraph (lines 814-817) sounds odd and a little out of place. How can knowledge on virulence and tropism help predict future outbreaks? And how does climate change come into play? I understand what the authors are trying to say (impact on public health decisions, early detection through sequencing etc.), but this should either be expanded, deleted or an alternate concluding paragraph should be considered.

Minor comments

  • Lines 555-557: Please delete (haws,.) or modify, and delete/replace the multiple “etc.” within this paraph.
  • Lines 636-637: Consider adding here (or anywhere in the avian model section) that the host competence of birds is to be considered when evaluating fitness and virulence. There are bird species with low, moderate and high susceptibility to WNV and evaluation of virulence factors in vivo will be influenced by the choice species to some extent. Also, the geographic distribution and abundance of avian species that have different susceptibility will influence the amplification and transmission dynamics locally.

Author Response

Our response to the reviewer's comments appear in bold+italics

Summary

The review manuscript “Molecular determinants of West Nile virus virulence and pathogenesis in vertebrate and invertebrate hosts” by Fiacre and colleagues is well written with an appropriate balance between in-text data/information and graphical presentation. The individual sections are comprehensive, yet succinctly summarized and the body of literature cited is adequate.

Many thanks for reviewing our manuscript. We considered your major and minor comments and modified the manuscripts accordingly, as indicated below. 

Major comments

  • Please double check the entire text for proper use of grammar. The majority of the text is written in good English, but there are certain paragraphs (the abstract is one of them) that would benefit from a grammar check. The manuscript has been reviewed by an English native speaker before resubmission.
  • The concluding paragraph (lines 814-817) sounds odd and a little out of place. How can knowledge on virulence and tropism help predict future outbreaks? And how does climate change come into play? I understand what the authors are trying to say (impact on public health decisions, early detection through sequencing etc.), but this should either be expanded, deleted or an alternate concluding paragraph should be considered. The paragraph was rewritten and completed in order to explicitly address how studies on virus genetic determinants could help in the control of WNV outbreaks. Lines 971-985.

Minor comments

  • Lines 555-557: Please delete (haws,.) or modify, and delete/replace the multiple “etc.” within this paraph. Corrections brought (lines 665-666).
  • Lines 636-637: Consider adding here (or anywhere in the avian model section) that the host competence of birds is to be considered when evaluating fitness and virulence. There are bird species with low, moderate and high susceptibility to WNV and evaluation of virulence factors in vivo will be influenced by the choice species to some extent. Also, the geographic distribution and abundance of avian species that have different susceptibility will influence the amplification and transmission dynamics locally. This information was added in the manuscript, lines 675-680 “Host competence should be considered when evaluating WNV fitness and virulence in birds. Bird species with low (chicken), moderate and high (AMCRs, HOSPs) competence will differentially replicate WNV and consequently, evaluation of virulence factors in vivo will be influenced to some extent by the bird species used. Also, the geographic distribution and abundance of avian species will influence the amplification and transmission dynamics locally.”

Reviewer 2 Report

MAJOR COMMENTS

In this review, “Molecular determinants of West Nile virus virulence and pathogenesis in vertebrate and invertebrate hosts,” Fiacre et al. discuss the the known molecular determinants of WNV virulence according to infected invertebrate (mosquitoes) or vertebrate hosts (mammalian and avian). This is a worthy topic, and the strengths of the manuscript include a well defined scope and very nice figures. Most of the information required for this review is already included in the manuscript. However, as currently written, the manuscript is somewhat rambling and difficult to follow. The introduction lacks key details (see below), particularly regarding the prevalence of WNV infection and disease in various hosts. More importantly, the specific sections on the virulence factors within each WNV gene are literally just a list of studies performed. I believe these latter sections need to be completely reworked. Instead of describing the details of experiments in list-form, the authors need to do a better job synthesizing this information into a coherent understanding of virulence for each gene, focusing first on the complete gene-product, then narrowing in on particular regions of the protein/gene of interest based on known function. How do mutations/glycosylation/etc affect this function? How might this relate to pathogenesis? Identify areas where this is unknown and could be a fruitful area of future research. Additionally, there is very little discussion of the immunogenicity (and subsequent immunopathology) that may play a role in WNV pathogenesis. The authors also need to link the specific determinants of virulence described in later sections to the broader themes in pathogenesis described in the introduction. 

MINOR COMMENTS

  • The manuscript would greatly benefit from additional editing by a native English speaker. 
  • Abstract: “3-5 horse vaccines are used”... which one is it? 3 or 5?
  • “Leading cause of neuroinvasive disease” needs a reference, I also don’t think this is true based on the general definition of “neuroinvasive disease.”
  • In many places, the Authors need to be more specific. For example, line 59 talks about lineage 2 WNV causing “mild fever with no impairment of the CNS”... but they fail to identify which host they are referring to. This is particularly important given that the topic of the review is comparison of WNV pathogenesis across hosts. 
  • Figure 1: “fecal-oral” is a more common term. 
  • Line 120: skin infection biology in mammals needs to be more thoroughly described. What about langerhans cells? 
  • “Retrograde axonal transport”... needs citation. Also, there is no nuanced discussion of how common each of these mechanisms for crossing the BBB are, and which one is primarily responsible for neuroinvasive disease in humans. 
  • The discussion on quasispecies is grossly inadequate. All viruses exist as quasispecies; this is a result of error-prone replication AND the subsequent influence of selective pressures. This is not unique to WNV. What selective pressures result in greater diversity in mosquito hosts vs. mammalian hosts? How do the population bottlenecks during the different modes of transmission depicted in Fig 1 impact diversity?
  • Lines 181-203 are completely outside the scope of this review and should be removed. 
  • Some statements are bizarre and inaccurate. For example, line 237... C57BL/6 mice are “fragile and difficult to breed.” 

Author Response

Our responses to the reviewer's comments appear in bold+italics

MAJOR COMMENTS

In this review, “Molecular determinants of West Nile virus virulence and pathogenesis in vertebrate and invertebrate hosts,” Fiacre et al. discuss the the known molecular determinants of WNV virulence according to infected invertebrate (mosquitoes) or vertebrate hosts (mammalian and avian). This is a worthy topic, and the strengths of the manuscript include a well defined scope and very nice figures. Most of the information required for this review is already included in the manuscript. However, as currently written, the manuscript is somewhat rambling and difficult to follow. The introduction lacks key details (see below), particularly regarding the prevalence of WNV infection and disease in various hosts. More importantly, the specific sections on the virulence factors within each WNV gene are literally just a list of studies performed. I believe these latter sections need to be completely reworked. Instead of describing the details of experiments in list-form, the authors need to do a better job synthesizing this information into a coherent understanding of virulence for each gene, focusing first on the complete gene-product, then narrowing in on particular regions of the protein/gene of interest based on known function. How do mutations/glycosylation/etc affect this function? How might this relate to pathogenesis? Identify areas where this is unknown and could be a fruitful area of future research. Additionally, there is very little discussion of the immunogenicity (and subsequent immunopathology) that may play a role in WNV pathogenesis. The authors also need to link the specific determinants of virulence described in later sections to the broader themes in pathogenesis described in the introduction. 

Many thanks for reviewing our manuscript and for your helpful comments. Information on WNV infections and disease in vertebrate hosts was completed (lines 124-133 for WNV infections in birds while lines 97-103 present WNV disease in humans). Prevalence of infection is highly variable, depending on the vertebrate species and geographic areas considered, WNV isolates and virulence properties, the co-circulation of other flaviviruses that could confer partial protection against WNV disease,…

The immunopathogenesis of WNV infection has been described lines 79-106 and although highly relevant for WNV pathogenesis and virulence, the majority of studies on WNV virulence determinants did not include histological or immune assays, precluding the analysis of the impact of viral genetic determinants on the immune response against WNV.

In most studies on WNV virulence molecular determinants, modifications brought to enzymatic and structural functions or to protein-protein interactions that are key for protein functions have not been evaluated and we feel that we presented current knowledge on WNV virulence determinants, describing the functions established for each viral protein and gathering experimental data and conclusions on every genetic determinants.

MINOR COMMENTS

  • The manuscript would greatly benefit from additional editing by a native English speaker. The manuscript has been reviewed by an English native speaker before resubmission.

  • Abstract: “3-5 horse vaccines are used”... which one is it? 3 or 5? 3 vaccines are used in Europe, while 5 are available in North America. We modified the corresponding sentence, line 21-22. “Several vaccines are used in horses in different parts of the world,…”
  • “Leading cause of neuroinvasive disease” needs a reference, I also don’t think this is true based on the general definition of “neuroinvasive disease.” A reference was added line 35.
  • In many places, the Authors need to be more specific. For example, line 59 talks about lineage 2 WNV causing “mild fever with no impairment of the CNS”... but they fail to identify which host they are referring to. This is particularly important given that the topic of the review is comparison of WNV pathogenesis across hosts. The host was specified at lines 63-64.
  • Figure 1: “fecal-oral” is a more common term. Figure 1 was modified accordingly.
  • Line 120: skin infection biology in mammals needs to be more thoroughly described. What about langerhans cells? Additional information on WNV infection in the skin has been added, lines 160-165.
  • “Retrograde axonal transport”... needs citation. Also, there is no nuanced discussion of how common each of these mechanisms for crossing the BBB are, and which one is primarily responsible for neuroinvasive disease in humans. The reference was added line 199 and the importance of the mechanisms involved in BBB crossing was discussed, lines 176-183.
  • The discussion on quasispecies is grossly inadequate. All viruses exist as quasispecies; this is a result of error-prone replication AND the subsequent influence of selective pressures. This is not unique to WNV. What selective pressures result in greater diversity in mosquito hosts vs. mammalian hosts? How do the population bottlenecks during the different modes of transmission depicted in Fig 1 impact diversity? The following sentences were added in the text, lines 214-219 “This results from several bottlenecks that shape the virus divergence during the progress of infection in birds, including the host defenses, varying cellular environments in different tissues and anatomic barriers such as the BBB. These multiple selective pressures determine tissue tropism and virus pathogenesis.” Bird species considered in WNV evolution studies were generally poorly competent for WNV replication (chicken in particular), mechanically restricting WNV genetic diversity. Host and vector competence levels are major selective bottlenecks for WNV and arboviruses in general.
  • Lines 181-203 are completely outside the scope of this review and should be removed. We think instead that the evolution of techniques enabling the construction of WNV reverse genetics tools provides technological breakthrough that support a more rapid characterization of WNV genetic determinants nowadays. For this reason, we believe that the corresponding section should be maintained in the new version.
  • Some statements are bizarre and inaccurate. For example, line 237... C57BL/6 mice are “fragile and difficult to breed.” The corresponding sentence was deleted, line 313.

Round 2

Reviewer 2 Report

The authors have addressed my concerns.